# PeerJ

# Identification of miR-194-5p as a potential biomarker for postmenopausal osteoporosis

Jia Meng[1,5], Dapeng Zhang[2,5], Nanan Pan[1], Ning Sun[3], Qiujun Wang[1], Jingxue Fan[1], Ping Zhou[1], Wenliang Zhu[4] and Lihong Jiang[1]

[1] Department of Geriatrics, The Second Affiliated Hospital of Harbin Medical University, Harbin, China
[2] Department of Orthopedic Surgery, The Fourth Affiliated Hospital of Harbin Medical University, Harbin, China
[3] Department of Nursing, The Second Affiliated Hospital of Harbin Medical University, Harbin, China
[4] Institute of Clinical Pharmacology, The Second Affiliated Hospital of Harbin Medical University, Harbin, China
[5] These authors contributed equally to this work.

Corresponding authors
Wenliang Zhu, wenzwl@yeah.net
Lihong Jiang,
jianglihong2006@163.com

## ABSTRACT

The incidence of osteoporosis is high in postmenopausal women due to altered estrogen levels and continuous calcium loss that occurs with aging. Recent studies have shown that microRNAs (miRNAs) are involved in the development of osteoporosis. These miRNAs may be used as potential biomarkers to identify women at a high risk for developing the disease. In this study, whole blood samples were collected from 48 postmenopausal Chinese women with osteopenia or osteoporosis and pooled into six groups according to individual T-scores. A miRNA microarray analysis was performed on pooled blood samples to identify potential miRNA biomarkers for postmenopausal osteoporosis. Five miRNAs (miR-130b-3p, -151a-3p, -151b, -194-5p, and -590-5p) were identified in the microarray analysis. These dysregulated miRNAs were subjected to a pathway analysis investigating whether they were involved in regulating osteoporosis-related pathways. Among them, only miR-194-5p was enriched in multiple osteoporosis-related pathways. Enhanced miR-194-5p expression in women with osteoporosis was confirmed by quantitative reverse transcription–polymerase chain reaction analysis. For external validation, a significant correlation between the expression of miR-194-5p and T-scores was found in an independent patient collection comprised of 24 postmenopausal women with normal bone mineral density, 30 postmenopausal women with osteopenia, and 32 postmenopausal women with osteoporosis ($p < 0.05$). Taken together, the present findings suggest that miR-194-5p may be a viable miRNA biomarker for postmenopausal osteoporosis.

## INTRODUCTION

Postmenopausal women have a high incidence of osteoporosis due to simultaneous existence of multiple independent predisposing factors, such as estrogen deficiency,

continuous calcium loss, and aging (*Rosen, 2005*). Osteoporosis is mainly characterized by an imbalance between excessive bone resorption by osteoclasts and bone formation by osteoblasts (*Raisz, 2005*). Osteoporosis leads to low bone mineral density (BMD) and decreased bone strength, which leads to increased risk of fragility fractures (*Kanis, 2002*). Osteoporosis is a major public health concern worldwide, especially in countries with large aging populations, such as China (*Feng et al., 2012*). In contrast to developing better treatments for patients with osteoporosis, a proactive approach that identifies patients at high risk for developing osteoporosis is recommended to prevent bone loss, especially when the number of patients with osteoporosis is estimated to grow (*Sanders & Geraci, 2013*; *Tella & Gallagher, 2014*). This viewpoint has prompted researchers to develop new osteoporotic biomarkers in recent years to identify postmenopausal women who are at high risk for developing osteoporosis (*Vasikaran et al., 2011*; *Garnero, 2014*).

MicroRNAs (miRNAs) are a superfamily of small (∼22 nucleotides), single-stranded, non-coding RNAs (*Bushati & Cohen, 2007*). Numerous studies have suggested that miRNAs are important regulators of bone metabolism, and that they participate extensively in bone formation and resorption (*Lian et al., 2012*; *Zhao et al., 2014*). For instance, miR-2861 was found to play an important role in promoting osteoblast differentiation by targeting histone deacetylase 5, whereas suppression of miR-2861 led to primary osteoporosis (*Li et al., 2009*). In addition, a critical miRNA regulator of osteoclast differentiation, miR-223, was also identified (*Sugatani & Hruska, 2007*). Because of its implications in osteoclast differentiation, miR-223 may be a promising therapeutic target for correcting excessive osteoclast-mediated bone resorption, which contributes to the pathological development of osteoporosis (*Raisz, 2005*).

Substantial evidence has shown that circulating miRNAs can be used as potential biomarkers for various human diseases (*Mitchell et al., 2008*; *Häusler et al., 2010*; *Taurino et al., 2010*; *Ries et al., 2014*). Just recently, studies suggested that three miRNAs, miR-21, miR-133a, and miR-422a, could be feasible biomarkers for postmenopausal osteoporosis (*Wang et al., 2012*; *Cao et al., 2014*; *Li et al., 2014*). In this study, we aimed to investigate whether other potential miRNA biomarkers exist. To do this, we conducted a microarray-based miRNA analysis in whole blood to search for circulating miRNAs with significantly altered peripheral blood expression profiles in postmenopausal osteoporotic women compared to postmenopausal women with osteopenia. T-score-based patient subgrouping was designed to identify miRNA biomarkers with altered expression profiles in peripheral blood that were correlated with a decline in BMD. A systematic pathway analysis was also performed using the Kyoto Encyclopedia of Genes and Genomes (KEGG) (*Kanehisa et al., 2006*), to reveal whether functional association exists between potential miRNA biomarkers and osteoporosis-related pathways. Lastly, quantitative reverse transcription-polymerase chain reaction (qRT-PCR) was used to validate independently the potential biomarkers identified from the initial screen. In summary, our efforts aim to provide new miRNA biomarkers as tools allowing for better diagnosis and prevention of postmenopausal osteoporosis in the future.

**Table 1 Characteristics of participants recruited for the microarray study.**

| Disease ($n$) | Subgroup ($n$) | Age (year) | Spine T-score | Femoral neck T-score |
|---|---|---|---|---|
| Osteopenia ($n = 23$) | S1, $-1.5 \leq$ Spine T-score $\leq -1.0$ ($n = 7$) | $63.1 \pm 2.4$ | $-1.21 \pm 0.20$ | $-0.99 \pm 0.63$ |
| | S2, $-2.0 \leq$ Spine T-score $< -1.5$ ($n = 7$) | $66.4 \pm 3.4$ | $-1.84 \pm 0.18$ | $-1.24 \pm 1.03$ |
| | S3, $-2.5 <$ Spine T-score $< -2.0$ ($n = 9$) | $64.8 \pm 3.7$ | $-2.19 \pm 0.12$ | $-1.31 \pm 0.60$ |
| Osteoporosis ($n = 25$) | S4, $-3.0 \leq$ Spine T-score $\leq -2.5$ ($n = 10$) | $66.3 \pm 3.6$ | $-2.79 \pm 0.14$ | $-2.27 \pm 0.76$ |
| | S5, $-4.0 \leq$ Spine T-score $< -3.0$ ($n = 9$) | $64.6 \pm 3.5$ | $-3.53 \pm 0.28$ | $-2.24 \pm 0.69$ |
| | S6, Spine T-score $< -4.0$ ($n = 6$) | $68.0 \pm 2.0$ | $-4.72 \pm 0.42$ | $-2.77 \pm 0.54$ |

Notes.
Data are expressed as the mean $\pm$ SD. There was no significant age difference across participant subgroups.

## MATERIALS AND METHODS

### Participant characteristics

This study was approved by the Ethics Committee of Harbin Medical University (Approval number: 2014-R-020). Each participant enrolled in this study was informed of the project and had signed a written consent. Twenty-three postmenopausal Chinese women with osteopenia (moderate decrease in BMD, spine T-score $\leq -1.0$ and $> -2.5$) and 25 postmenopausal Chinese women with osteoporosis (marked decline in BMD, spine T-score $\leq -2.5$) were recruited and further divided into six subgroups according to their T-score measurements (Table 1). Dividing the patients into six subgroups allowed investigation of whether altered expression of a particular miRNA may be correlated with bone loss during pathological progression from osteopenia to osteoporosis. Both the spine and femoral neck T-score was measured for each participant. In addition, an independent collection of participants was recruited for the purpose of external validation. This group was comprised of 24 postmenopausal women with a normal BMD, 30 postmenopausal women with osteopenia, and 32 postmenopausal women with osteoporosis (Table S1). Participants were given a questionnaire to obtain their medication history, and a hospital examination verified that no serious complications, such as cancer, Type II diabetes, and/or cardiac disease were present. All the participants ranged in age from 59 to 70 and had been postmenopausal for at least 12 months (no menses 12 months after the date of the last menses). T-scores of the spine (L1-L4) and femoral neck were obtained using a dual energy X-ray absorptiometry scanner (Hologic Inc., Bedford, Massachusetts, USA).

### Blood sample collection and total RNA extraction

Five milliliters of whole blood was obtained from each participant. Each whole blood sample was independently lysed using Red Blood Cell (RBC) Lysis Solution (Beyotime, Shanghai, China) and centrifuged for 10 min at $450 \times$ g. TRIzol reagent (Invitrogen, Shanghai, China) was used to extract RNA from the precipitate. RNA extraction was completed within 30 min after blood collection from each participant. Isolated RNA eluate was stored at $-80$ °C. After blood sample collection was complete, all RNA extraction samples from individual participants were thawed and pooled separately into six subgroups corresponding to the spine T-score values (Table 1). The pooled samples

were then stored at $-80\,^{\circ}\text{C}$ for future experiments. The same procedure was applied to the whole blood samples for external validation (Table S1); however, these RNA samples were not pooled.

## Microarray scanning

An Agilent Human miRNA microarray (Release 19.0, $8 \times 60$ K) was used for global scanning of miRNA expression in pooled RNA samples. Sample labeling, microarray hybridization, and washing were performed based on the manufacturer's standard protocols (Agilent Technologies Inc., Santa Clara, California, USA). Briefly, total RNA was dephosphorylated, denatured, and then labeled with Cyanine-3-CTP. After purification, labeled RNAs were hybridized onto the microarray. After washing, the arrays were scanned with an Agilent Scanner G2505C (Agilent Technologies Inc., Santa Clara, California, USA). Feature Extraction software (version 10.7.1.1; Agilent Technologies Inc., Santa Clara, California, USA) was used to analyze microarray images and obtain raw data. Next, GeneSpring software (version 12.5; Agilent Technologies Inc., Santa Clara, California, USA) was used to complete the basic analysis using raw data. The raw data was normalized with the quantile algorithm. If the probes with a positive normalized expression value were flagged as "Detected" in at least 100% of samples, they were chosen for further analysis. Differentially expressed miRNAs were then identified through fold change as well as the $p$ value calculated using a Student's $t$-test. The threshold set for significantly up- and down-regulated genes was a fold change $>2.0$ and a $p$ value $<0.05$. The miRNA microarray assay was performed by Shanghai OEBiotech Technology Co, Ltd. (Shanghai, China). Microarray scanning data have been submitted to the Gene Expression Omnibus with an accession number GSE64433.

## qRT-PCR analysis

To avoid false positives during microarray detection, a qRT-PCR assay was used to validate further the significantly dysregulated miRNAs identified in the pooled RNA extraction samples. RNA eluate (0.4 µL) was reverse transcribed into cDNA in a 10-µL reaction volume, using a high-capacity cDNA reverse transcription kit (Applied Biosystems, Foster City, CA). Briefly, 1.0 µL $10 \times$ RT Buffer, 0.4 µL $25 \times$ dNTP Mix (100 mM), 1.0 µL 100 mM RT miRNA primers, 0.5 µL MultiScribe[TM] Reverse Transcriptase, and 0.5 µg RNA were mixed, and the reaction volume was brought up to 10 µL by using DEPC $H_2O$. Transcription parameters were as follows: $25\,^{\circ}\text{C}$ for 10 min, $37\,^{\circ}\text{C}$ for 120 min, $85\,^{\circ}\text{C}$ for 5 min. The cDNA was diluted 1:100 and assayed in 10 µL PCR reactions. miRNA primer sequences and the U6 internal control are listed in Table S2. For real-time PCR, 10 µL $2 \times$ SYBR Green PCR Master Mix (Applied Biosystems, Warrington, UK), 7 µL DEPC $H_2O$, 1 µg cDNA, and 1 µL (each) F and R primers were mixed and run on a Bio-Rad CFX96 Touch[TM] Real-Time PCR Detection System (Bio-Rad, Singapore). Samples were briefly heated (10 min, $95\,^{\circ}\text{C}$) and amplified as follows: $95\,^{\circ}\text{C}$ for 15 s and $60\,^{\circ}\text{C}$ for 1 min for 40 cycles. MiRNA cycle threshold (CT) values were normalized to U6 and calculated using the equation $2^{-\Delta\Delta\text{CT}}$, as previously described (Wang et al., 2012). The same procedure was applied to the RNA samples from the external validation set (Table S1).

Peer J

Additionally, the efficiency and specificity of the qRT-PCR protocol was evaluated for the miR-194-5p primers designed for this study. A synthetic miR-194-5p was generated according to its nucleotide sequence in miRBase (*Kozomara & Griffiths-Jones, 2014*). Synthetic miR-194-5p was dissolved using DEPC $H_2O$ at a concentration of 10 μg/μL. The solution was then further diluted $4 \times 10^3$, $4 \times 10^4$, $4 \times 10^5$, and $4 \times 10^6$ times. After this, 1 μL of each solution at different concentrations was used as a template to be co-amplified with the cDNA reverse-transcribed from the RNA obtained from blood samples and be subjected to real-time PCR assay.

### Pathway analysis

The HuGE Navigator Gene Prospector online tool was used to search for literature-reported osteoporosis-related genes (*Yu et al., 2008*). After uploading the official symbols of osteoporosis-related genes onto the DAVID website, the online functional annotation tool was applied to identify osteoporosis-related pathways by selecting 'Homo sapiens' as the species background and 'KEGG_pathway' as the only annotation for functional analysis (*Huang da, Sherman & Lempicki, 2009*). Over-representation of osteoporosis-related genes on a KEGG pathway was considered statistically significant only if the Bonferroni-adjusted $p < 0.05$ (*Bland & Altman, 1995*).

An integrative retrieval from two miRNA-target interaction (MTI) databases, including miRSel (*Naeem et al., 2010*) and miRTarBase (*Hsu et al., 2014*), was applied to search for experimentally validated target genes of six miRNAs, including miR-130b-3p, -133a, -151a-3p, -151b, -194-5p, and -590-5p in humans. After uploading the official symbols of the target genes of each miRNA onto the DAVID website, the online functional annotation tool was used to reveal whether these target genes were involved in osteoporosis-related KEGG pathways.

### Statistical analysis

All data are expressed as the mean $\pm$ SD. Statistical analysis was performed using Student's $t$-test or a Pearson correlation test. GraphPad Prism v6.0 was used to conduct statistical analysis. Differences were considered as statistically significant when $p < 0.05$.

## RESULTS

### Microarray scanning identified six miRNAs with increased peripheral blood expression in participants with osteoporosis

To search for potential miRNA biomarkers for postmenopausal osteoporosis, a comprehensive miRNA expression analysis was performed on pooled RNA samples isolated from postmenopausal Chinese women with osteopenia or osteoporosis (Table 1). We identified 331 unique miRNAs with detectable expression in every sample tested (Table S3). Among them, six miRNAs (miR-130b-3p, -151a-3p, -151b, -194-5p, -590-5p, and -660-5p) were found to have significantly increased peripheral expression in the blood of participants with osteoporosis compared those with osteopenia ($p < 0.05$, Fig. 1). MiR-194-5p was the most highly upregulated (with a more than 5-fold change), followed by miR-151a-3p,

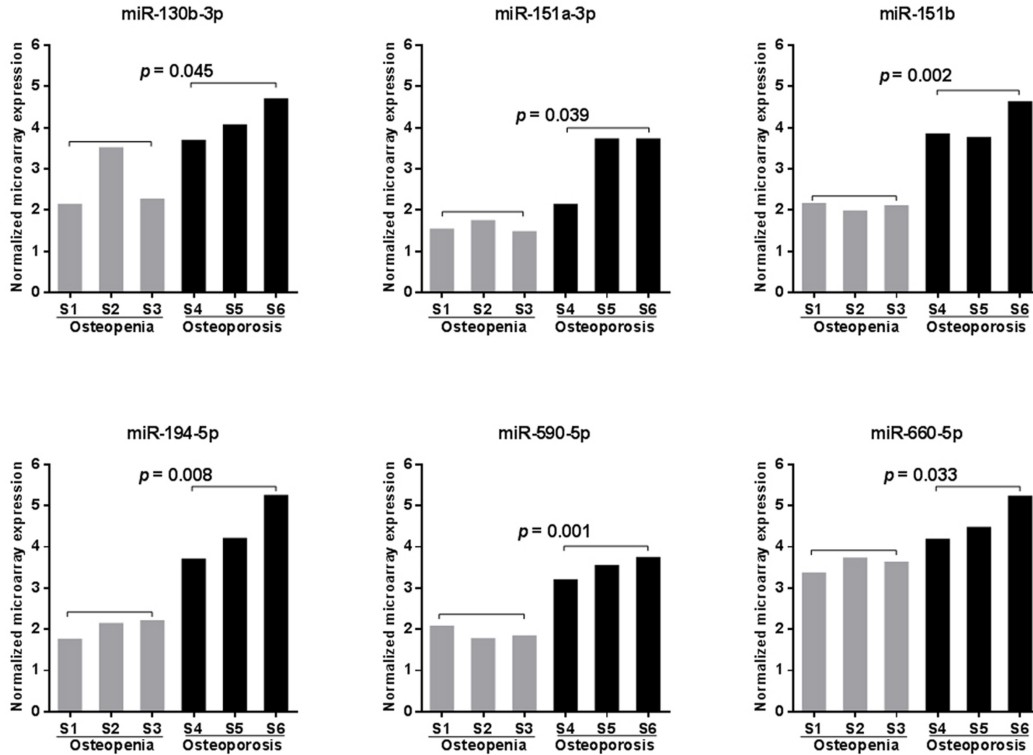

**Figure 1 Microarray scanning identified six significantly upregulated miRNAs in samples obtained from patients with postmenopausal osteoporosis.** Statistical comparison was performed between participants with osteopenia (S1–S3) and osteoporosis (S4–S6) using a Student's $t$-test.

-151b, and -590-5p (with a more than 3-fold change), and miR-130b-3p and -660-5p (with only a 2-fold increase in expression) (Table S3).

## qRT-PCR assay validated significant upregulation of five miRNAs in the blood of patients with osteoporosis

Due to the high false positive rate of the microarray method, we validated the circulating blood expression levels of the six significant microarray-identified miRNAs by using qRT-PCR. Except for miR-660-5p, all of the miRNAs tested showed remarkably higher expression in the peripheral blood of patients with osteoporosis compared to that observed in the blood of patients with osteopenia ($p < 0.05$, Fig. 2). Furthermore, we investigated whether a significant linear correlation existed between miRNA expression and spine T-score. Our results showed that the expression of four miRNAs, including miR-130b-3p, -151a-3p, -151b, -194-5p, in circulating blood were significantly and negatively correlated with a decline in BMD ($p < 0.05$, Fig. S1). Of note, the expression levels of miR-151b and -194-5p were also significantly and negatively correlated with femoral neck T-scores ($p < 0.05$, Fig. S1). This finding suggested that these two miRNAs might be more suitable for further consideration as potential biomarkers for postmenopausal osteoporosis.

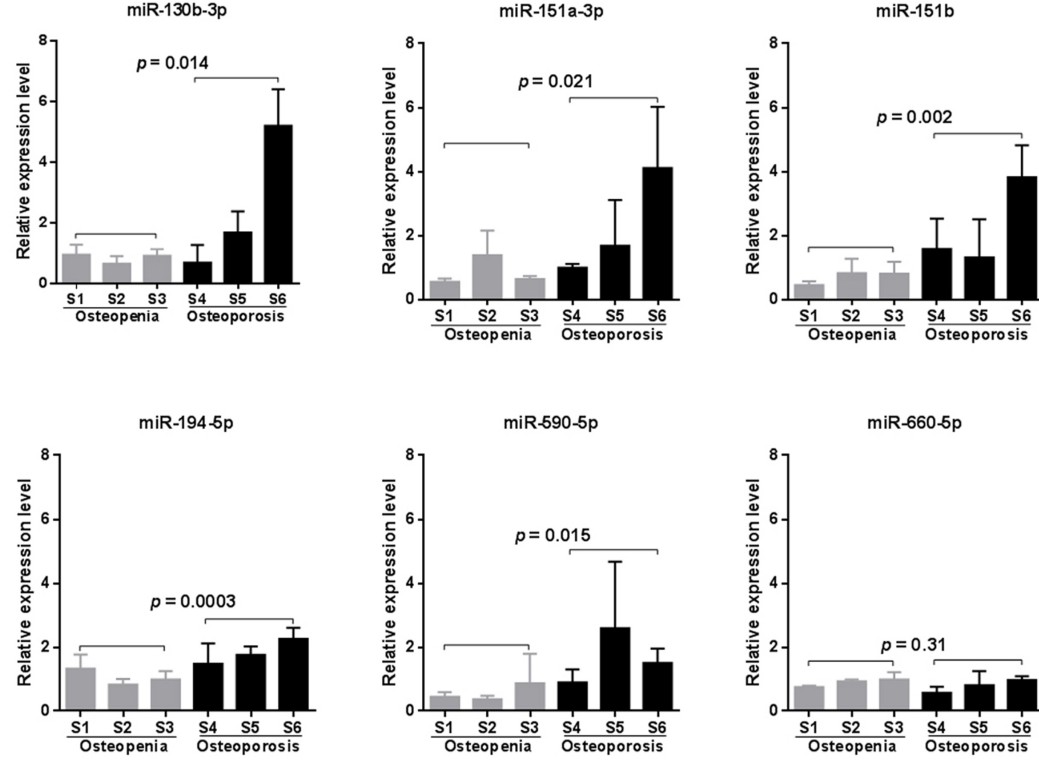

**Figure 2 qRT-PCR validation results ($n = 4$).** Statistical comparison was performed between participants with osteopenia (S1–S3) and those with osteoporosis (S4–S6), using a Student's $t$-test.

## miR-194-5p is implicated in multiple osteoporosis-related pathways

To explore a potential functional association between the miRNAs identified in this study and osteoporosis, the online bioinformatics tool DAVID was used to investigate whether the identified miRNAs regulate osteoporosis-related pathways by targeting the mRNA genes in each pathway. There were 985 genes reported in the literature to be functionally associated with osteoporosis. Analysis of their official gene symbols with the DAVID website revealed 19 KEGG pathways with over-represented osteoporosis-related genes (Table 2). These pathways were defined as osteoporosis-related due to this significantly enrichment with osteoporosis-related genes. Experimental evidence of MTIs were retrieved from the miRSel and miRTarBase databases for the five PCR-validated miRNAs (miR-130b-3p, -151a-3p, -151b, -194-5p, -590-5p) identified in this study, as well as miR-133a,which was previously suggested as a feasible biomarker for postmenopausal osteoporosis (*Wang et al., 2012*; *Li et al., 2014*). We then investigated involvement of each miRNAs in regulating osteoporosis-related pathways. MiR-151b was found to lack experimentally-validated MTIs; therefore, it was excluded from further analysis. The remaining four miRNAs and miR-133a, the known osteoporosis biomarker, underwent subsequent pathway analysis. As expected, miR-133a was implicated in six osteoporosis-related pathways such as *cytokine-cytokine receptor interaction* (Table 2). This finding was consistent with

**Table 2  Results of KEGG pathway analysis.**

| KEGG pathway | Count (Adjusted $p$-value) | miRNA ($n$) |
| --- | --- | --- |
| Cytokine-cytokine receptor interaction | 84 (1.46E-17) | miR-133a (6) |
| Adipocytokine signaling pathway | 38 (2.84E-16) | – |
| TGF-beta signaling pathway | 38 (1.70E-11) | miR-130b-3p (4), miR-194-5p (6[a]), miR-590-5p (3) |
| Wnt signaling pathway | 50 (3.81E-10) | miR-194-5p (7[a]) |
| Steroid hormone biosynthesis | 25 (1.90E-9) | – |
| MAPK signaling pathway | 69 (6.66E-9) | miR-133a (7); miR-194-5p (6) |
| Apoptosis | 33 (7.94E-8) | miR-133a (4); miR-194-5p (3) |
| Hedgehog signaling pathway | 25 (3.42E-7) | – |
| Toll-like receptor signaling pathway | 33 (5.50E-6) | miR-194-5p (4) |
| Jak-STAT signaling pathway | 41 (6.06E-5) | miR-133a (6); miR-194-5p (6) |
| Hematopoietic cell lineage | 28 (8.20E-5) | – |
| Neurotrophin signaling pathway | 35 (1.02E-4) | miR-133a (4); miR-194-5p (4) |
| Androgen and estrogen metabolism | 16 (9.00E-4) | – |
| NOD-like receptor signaling pathway | 21 (1.61E-3) | – |
| T cell receptor signaling pathway | 28 (9.15E-3) | miR-133a (4); miR-194-5p (3) |
| Neuroactive ligand–receptor interaction | 51 (1.26E-02) | – |
| Complement and coagulation cascades | 20 (2.98E-2) | – |
| PPAR signaling pathway | 20 (2.98E-2) | – |
| RIG-I-like receptor signaling pathway | 20 (4.44E-02) | – |

**Notes.**

Osteoporosis-related genes in each KEGG pathway were counted if the Bonferroni-adjusted $p$-value was calculated to be less than 0.05. $n$, the number of pathway-related miRNA target genes. The short bar indicates that there is no miRNA to target any gene involved in the corresponding KEGG pathway.

[a] Indicates significantly enriched regulation of pathway-related genes (Bonferroni-adjusted $p < 0.05$).

the previous study, in which the authors demonstrated that miR-133a targeted multiple pathway-associated genes, such as chemokine (C-X-C motif) ligand 11 (CXCL11) and chemokine (C-C motif) receptor 3 (CXCR3) (*Wang et al., 2012*). In our analysis, we determined that miR-194-5p was functionally associated with eight osteoporosis-related pathways. This result implies that by targeting many genes, miR-194-5p may have effects on the TGF-beta and Wnt signaling pathways, which were shown to play critical roles in the pathology of postmenopausal osteoporosis (*Krishnan, Bryant & Macdougald, 2006*; *Nistala et al., 2010*).

## External validation confirmed increased expression of miR-194-5p in postmenopausal osteoporosis

External validation of miR-194-5p expression in the blood circulation of postmenopausal women with osteoporosis was performed by qRT-PCR analysis. To do this, an independent collection of participants was recruited, which was comprised of 24 postmenopausal women with normal BMD, 30 postmenopausal women with osteopenia, and 32 postmenopausal women with osteoporosis (Table S1). A significant increase in peripheral expression of miR-194-5p was found in postmenopausal women with osteopenia or osteoporosis, compared to postmenopausal women with normal BMD ($p < 0.001$, Fig. 3A). A remarkable negative correlation was also found between miR-194-5p expression and T-scores ($p < 0.05$, Figs. 3B and 3C). Moreover, the qRT-PCR method

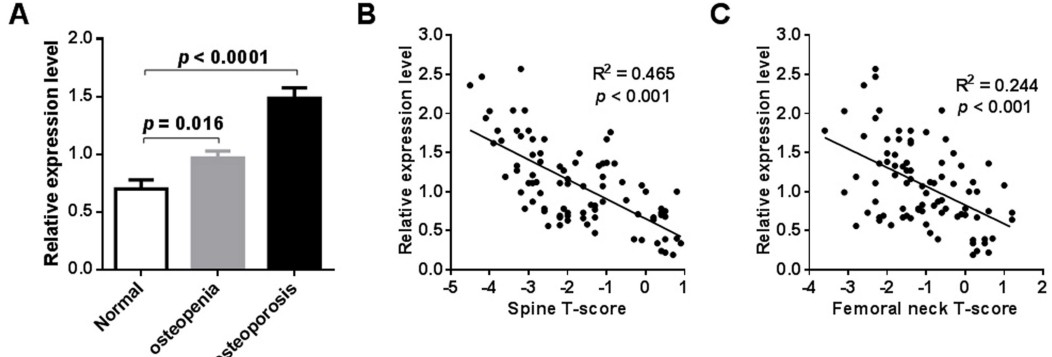

**Figure 3 Results of external validation by qRT-PCR analysis.** An obvious increase in miR-194-5p expression was observed in participants with osteopenia or osteoporosis compared with participants that had a normal BMD ($n = 86$) (A). There was no significant difference in age across he groups. A significant correlation was found between miR-194-5p expression and spine (B) and femoral neck T-scores (C).

established in this study efficiently detected miR-194-5p expression in peripheral blood with high specificity (Fig. S2), thus increasing our confidence in the reliability of the external validation results.

## DISCUSSION

Osteoporosis is an independent factor that increases the risk of fragility fractures in postmenopausal women (*Kanterewicz et al., 2014*). In this study, a microarray-based scanning approach followed by qRT-PCR validation and pathway analysis was conducted to identify potential circulating miRNA biomarkers for postmenopausal osteoporosis. Our study revealed that miR-194-5p should be considered a potential biomarker for postmenopausal osteoporosis, in addition to three previously recognized miRNAs, miR-21, -133a and -422a (*Wang et al., 2012*; *Cao et al., 2014*; *Li et al., 2014*).

A miRNA microarray technique was used in this study for high-throughput detection of hundreds of miRNAs at a relatively low cost compared to gene sequencing. The success of this method in identifying potential miRNA biomarkers has been widely confirmed in many studies on various diseases (*Mitchell et al., 2008*; *Häusler et al., 2010*; *Taurino et al., 2010*; *Ries et al., 2014*). However, a major pitfall of this approach is a high false positive rate. Thus, qRT-PCR is generally performed to validate significant miRNAs identified by microarray. In contrast to previous studies (*Wang et al., 2012*; *Cao et al., 2014*; *Li et al., 2014*), we used pooled RNA samples according to T-score measurements, rather than those of individual participants. This experimental design aims to minimize individual, physical differences to reveal potential relationships between miRNA expression and a decline in BMD. Indeed, this approach led to the identification of six dysregulated miRNAs by microarray analysis. Among them, elevated expression miR-151b and miR-194-5p was validated in the blood of patients with osteoporosis, and we determined a remarkable negative correlation with enhanced miR-194-5p expression and bone loss in both the spine and femoral neck by qRT-PCR analysis. However, these results should be taken with caution because the sample size of each T-score subgroup was small. In the case

of miR-194-5p expression, we confirmed its negative correlation with the decline of BMD in a larger independent population including 86 subjects covering a wider BMD range. This finding suggested that miR-194-5p could be a potential miRNA biomarker for osteoporosis because its peripheral expression is associated with pathological bone loss in postmenopausal women.

To further investigate potential functional roles of these significant miRNAs in osteoporosis, an evidence-based pathway analysis was performed on target genes of miRNAs and osteoporosis-related genes at the pathway level using KEGG, a reference knowledge base of signaling pathways in humans (*Kanehisa et al., 2006*). Experimental and literature-reported evidence for MTIs (*Naeem et al., 2010*; *Hsu et al., 2014*) and gene-osteoporosis association (*Yu et al., 2008*) increased our trust in the analysis results. There were nearly 1,000 genes reported in the literature that were related to osteoporosis, suggesting complex pathological mechanisms underlying this disease (*Raisz, 2005*; *Rachner, Khosla & Hofbauer, 2011*). In total, 19 KEGG pathways were found to be significantly associated with osteoporosis (Bonferroni-adjusted $p < 0.05$). Of note, approximately 9% of osteoporosis-related genes were involved in the *cytokine-cytokine receptor interaction* pathway, indicating critical roles for cytokines and their corresponding receptors in the pathogenesis of osteoporosis (*Manolagas & Jilka, 1995*). This result supported previous findings obtained from a bibliometric network (*Sun et al., 2013*). Although different data resources were used, Sun and colleagues (*2013*) also observed that *cytokine-cytokine receptor interaction* might be the most osteoporosis-related-genes-enriched KEGG pathway. Obvious gene enrichment in the *Wnt signaling pathway* was also determined by our pathway analysis. This finding was in agreement with the notion that Wnt antagonists should be further developed for osteoporosis treatment (*Rawadi & Roman-Roman, 2005*; *Chan, Van Bezooijen & Löwik, 2007*).

MiRNA can influence a signaling pathway by targeting many pathway-related genes (*Kowarsch et al., 2011*). Our analysis highlighted miR-194-5p as a potential biomarker for osteoporosis, as it was found to target genes in eight osteoporosis-related pathways. It was not surprising that miR-133a, a validated biomarker for osteoporosis (*Wang et al., 2012*; *Li et al., 2014*), targeted genes in six osteoporosis-related pathways. Based on our previous studies of multi-pathway renoprotectants (*Xu et al., 2014*), enriched regulation of multiple pathways associated with osteoporosis may suggest a potential functional role for miR-194-5p in the pathogenesis of osteoporosis.

Just recently, it was confirmed that by targeting chicken ovalbumin upstream promoter-transcription factor II (COUP-TFII), miR-194-5p stimulated osteogenesis and inhibited adipogenesis if it was specifically overexpressed in mesenchymal stem cells (*Jeong et al., 2014*). COUP-TFII was sought as a critical regulator of mesenchymal stem cell fate, regulating stem cell differentiation into osteoblasts and adipocytes. This observation suggested that miR-194-5p might play an important role in osteoblast differentiation. Furthermore, modulation of Wnt signaling was implicated in COUP-TFII-mediated regulation on mesenchymal cell commitment and differentiation (*Xie et al., 2011*). The results of our pathway analysis are in line with those of this report, as we determined

that miR-194-5p might influence Wnt signaling by targeting genes in the Wnt signaling pathway. Further functional experiments are needed to validate these results. Recently, it was demonstrated that both elevated bone resorption and stimulated bone formation could be simultaneously observed in an ovariectomy model, despite uncoupling between excessive bone resorption by osteoclasts and bone formation by osteoblasts that eventually led to a net bone loss (*Matsuoka et al., 2014*). Taken together, we propose that the above evidences may reasonably explain the findings observed in our study where miR-194-5p showed elevated expression the peripheral blood of postmenopausal women with osteoporosis and, at the same time, a negative correlation between its expression and the extent of bone loss.

In conclusion, we integrated microarray-based miRNA scanning, qRT-PCR, and pathway analysis to identify potential miRNA biomarkers for postmenopausal osteoporosis. A limitation of our study was that only 331 miRNAs were detected by miRNA microarray. Despite this, we were able to identify miR-194-5p as a potential biomarker for postmenopausal osteoporosis. We confirmed that miR-194-5p expression was elevated in the peripheral blood of postmenopausal osteoporotic women. Furthermore, miR-194-5p was negatively correlated with the decline of BMD in the spine and femoral neck, and it was found to be associated with multiple osteoporosis-related pathways. Nevertheless, further research in a larger study population will be necessary to validate our findings.

### Funding

This work was supported by National Natural Science Foundation of China (No. 31301095) and Foundation of Health Department of Heilongjiang Province (No. 2010-213). The funders had no role in study design, data collection and analysis, decision to publish, or preparation of the manuscript.

### Grant Disclosures

The following grant information was disclosed by the authors:
National Natural Science Foundation of China: 31301095.
Foundation of Health Department of Heilongjiang Province: 2010-213.

### Competing Interests

The authors declare there are no competing interests.

### Author Contributions

- Jia Meng conceived and designed the experiments, performed the experiments, analyzed the data, wrote the paper, prepared figures and/or tables, reviewed drafts of the paper.
- Dapeng Zhang performed the experiments, analyzed the data, reviewed drafts of the paper.
- Nanan Pan, Ning Sun, Qiujun Wang, Jingxue Fan and Ping Zhou performed the experiments, reviewed drafts of the paper.

- Wenliang Zhu conceived and designed the experiments, contributed reagents/materials/analysis tools, wrote the paper, reviewed drafts of the paper.
- Lihong Jiang contributed reagents/materials/analysis tools, wrote the paper, reviewed drafts of the paper.

### Human Ethics

The following information was supplied relating to ethical approvals (i.e.,, approving body and any reference numbers):

This study was approved by the Ethics Committee of Harbin Medical University (Approval number: 2014-R-020). Each participant enrolled in this study was informed of the project and signed a written consent.

### Microarray Data Deposition

The following information was supplied regarding the deposition of microarray data:

GSE64433 / NCBI GEO: http://www.ncbi.nlm.nih.gov/geo/query/acc.cgi?acc=GSE64433.

### Supplemental Information

Supplemental information for this article can be found online at http://dx.doi.org/10.7717/peerj.971#supplemental-information.

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
