# Peer review of "Identification of miR-194-5p as a potential biomarker for postmenopausal osteoporosis"

_PeerJ, doi:10.7717/peerj.971_

## Round 0.1 · original submission · Major Revisions

Both reviewers feel that a more adequate methodological description is necessary, and these details should therefore been provided. Also, reviewer 2 requires miRNA studies in BMD normal (control) postmenopausal women to confer reliability to your findings - even preliminary data would be acceptable.

Instead, the detection of a limited number (331) of miRNAs by miRNA microarray assay in the six pooled RNA extraction samples could be discussed.

Reviewer 1 ·

Basic reporting

Please check typo errors through the manuscript (row 150/151, row 202)

Experimental design

1) The authors should better specify the procedure of pooling. Were RNA samples quantified by a spectrophotometer before the pooling? Which quantity of RNA was used to prepare the pooled samples ?
2) Since the primers used for miRNA quantification by qRT-PCR are designed by the authors, the efficiency and specificity of their qRT-PCR protocol should be evaluated. For instance, serial dilutions of a synthetic microRNA should be prepared and quantified by qRT-PCR to determine the protocol efficiency and the specificity of the reaction. This is mandatory at least for microRNAs with significant results.
3) Relevant details of the amplification protocol are lacking. Is it a stem-loop PCR adopted for miRNA quantifications ?

Validity of the findings

No comments

Additional comments

In this study the authors with the aim to find a molecular marker for osteoporosis show a significant correlation between quantification of miR-194 in blood cells and T-score in a sample of post-menopausal women with osteoporosis. The study, although preliminary, is potentially interesting but some technical questions deserve to be answered by the authors.

Reviewer 2 ·

Basic reporting

There were too many English grammar and vocabulary errors in this manuscript, which made it difficult to be understood. Please ask an English native speaker to edit the manuscript.

The paragraph on Lines 185-195 should be deleted because that paragraph was just repeated, which was the almost same as the paragraph on Line 172-184.

Experimental design

Appropriate controls such as age-matched healthy women should be used to compare with patients with osteoporosis or osteopenia regarding the differences of miRNA expression. Without appropriate controls, the results may not be reliable.

Validity of the findings

Please explain the principle for the qRT-PCR assay, which was used to validate the significantly dysregulated miRNAs identified by the microarray expression scanning on the pooled RNA extraction samples. It is hard to understand if the qRT-PCR validation method (including the high-capacity cDNA reverse transcription kit and miRNA primers, on Lines 139-145) described in the manuscript did work or not.

Additional comments

In the manuscript, totally 331 miRNAs were detected by miRNA microarray assay in all of the six pooled RNA extraction samples (Table S3). Usually, over 1000 miRNAs could be detected in human samples by the microarray hybridization assay. Thus the results presented in this manuscript may not reveal all of the dysregulated miRNAs expressed in the patients with osteoporosis.

Please keep the same names of miRNAs in the manuscript. For instance, miR-130b-3p was used in Table S3; however, miR-130b was shown in the main text and Figure 1.

On Line 184, the sentence “…… osteopenia and 32 postmenopausal women with osteoporosis (Table S1)” was changed into “…… osteopenia and 32 postmenopausal women with osteoporosis (Table S1 and Fig. S1).

On Line 314, the statement “In conclusion, we established a novel method to ……” is not appropriate. Microarray assay used in this manuscript was not a novel method to detect miRNA expression.

The legend for “Table S2 Sequences of the genes involved in this study (human)” should be corrected. This was because the primer sequences were described in Table 2.

In Figure 1, the error bars should be shown.

In Figure 3 and Figure S1, the values for the coefficient of determination (R2) should be presented to confirm the significant correlation between the measured values of miR-194 and Spine T-scores and Femoral neck T-scores.

On Lines 205 and 207, “Fig. S3” should be changed as “Fig. S1”.

---

## Round 0.2 · Minor Revisions

Although many issues raised by the reviewers have been addressed, there is still a methodological concern raised by reviewer 1, which needs to be addressed.

Reviewer 1 ·

Basic reporting

No Comments

Experimental design

In order to check sensitivity and specificity of their real-time protocol, the authors performed an experiment of serial dilution of synthetic miR-196 and subsequent quantification by real-time PCR. I feel that the protocol works but the experiment and the data presented by the authors are not reasonable (figure S2).
The concentrations chosen to demonstrate the sensitivity and the efficiency of the assay are too high. The lowest dilution used by the authors is approximately one thousand millions of copies. The experiment should be performed decreasing the amount of microRNA subjected to amplification. I am sure that starting from 10 million of copies and decreasing serially by a factor ten authors will get better results. In theory, every tenfold decrease of concentration should result in an increase of 3.33 cycles in the Ct, but values near this ideal number are acceptable.

Validity of the findings

No commentes

---

## Round 0.3 · accepted · Accept

The authors have fulfilled all the requests. In particular, experiment in Fig S2 has been repeated according to the reviewer's suggestions. In the same figure, "negtive" should be corrected with "negative".